# Evaluation of altered patterns of tactile sensation in the diagnosis and monitoring of leprosy using the Semmes-Weinstein monofilaments

**Marco Andrey Cipriani Frade** [1,2]*, **Fred Bernardes Filho** [1,2], **Claudia Maria Lincoln Silva** [1,2‡], **Glauber Voltan** [1,2], **Filipe Rocha Lima** [1,2], **Thania Loyola Cordeiro Abi-Rached** [1,2‡], **Natália Aparecida de Paula** [1,2]

1 Dermatology Division, Department of Clinical Medicine, Ribeirão Preto Medical School, University of São Paulo, Ribeirão Preto, Brazil, 2 National Reference Center in Sanitary Dermatology, Focusing on Leprosy, of Ribeirão Preto Clinical Hospital, Ribeirão Preto, Brazil

☯ These authors contributed equally to this work.
‡ CMLS and TLCAR also contributed equally to this work.
* mandrey@fmrp.usp.br

**Data Availability Statement:** All relevant data are within the paper and its Supporting Information files.

## Abstract

### Background

Leprosy neuropathy is the most common peripheral neuropathy of infectious etiology worldwide; it is characterized as asymmetric and focal multiple mononeuropathy. Semmes-Weinstein monofilament (SWM) test is a simple method to assess sensory nerve function.

### Methods and findings

In this prospective cohort study, a dermatologist carried out hands and feet tactile sensation test with SWM in 107 multibacillary leprosy patients at diagnosis and in 76 patients at the end of treatment from 2016 to 2019. At diagnosis, 81/107 (75.7%) patients had some degree of functional disability, and 46 (43%) of them had altered SWM-test in the hands and 94 (87.9%) of them in the feet. After one year of multibacillary multidrug therapy, the disability decreasing to 44/76 patients (57.9%) and decreasing of the percentual of patients with altered SWM-test to 18% for the hands, and to 28.7% for the feet. At the end of treatment, the number of SMW-test points presented improvement in the hands of 22 (28.9%) patients, and in the feet of 47 (61.8%). In the hands, by SWM-test, only the radial nerve point demonstrated a significant asymmetry, while in the feet, the difference between the sum of altered SWM-test points showed significant asymmetry between both sides, highlighting the tibial nerve for the establishment of asymmetric leprosy neuropathy. In Spearman's correlation analysis, a positive correlation with statistical significance was observed between the number of hands and feet SWM altered points at diagnosis and the degree of disability at diagnosis (0.69) and at the end of the treatment (0.80).

**Funding:** This study was supported by the WHO Implementation Research Team of Ribeirão Preto Medical School in the form of a grant awarded to MACF (771/2016 SCAPIR), the Center of National Reference in Sanitary Dermatology focusing on Leprosy of Ribeirão Preto Clinical Hospital, Ribeirão Preto, São Paulo, Brazil in the form of funds awarded to MACF, the Brazilian Health Ministry (MS/FAEPAFMRP-USP) in the form of grants awarded to MACF (749145/2010, 767202/2011), and Fiocruz Ribeirão Preto - TED 163/2019 in the form of a grant awarded to MACF (Processo: N˚ 25380.102201/2019-62/ Projeto Fiotec: PRES-009-FIO-20); and National Council for Scientific and Technological Development (CNPq) with Ph.D. scholarships program for FL and research grant for MACF (423635/2018-2)." All the funders had no role in study design, data collection and analysis, decision to publish, or preparation of the manuscript.

**Competing interests:** The authors have declared that no competing interests exist.

## Conclusion

The patterns of hands and feet tactile sensation at diagnosis and their consequent modifications with the anti-leprosy drugs define the bacterial etiology of neuropathy, an important tool for the clinical diagnosis and follow up of the disease, highlighting the tibial nerve findings, the most affected nerve among leprosy patients by SWM-test, with significant asymmetry and focality impairments.

## Introduction

Among the neuropathies caused by bacteria, there are Lyme disease and leprosy [1]. While in the first, peripheral neuropathy is usually a late, mild, diffuse and "sock and glove" distribution, in leprosy, neural involvement is early and characterized as asymmetric and focal multiple mononeuropathy [2, 3]. The improvement of neurological symptoms in leprosy with antibacterial drugs is evidence of the infectious cause of neuropathy, and it should be valued as therapeutic evidence, mainly for patients with negative complementary laboratory tests.

In leprosy, the existing data on neurological manifestations and the predominance of sensory or motor involvement are often contradictory and the methodologies and objectives of the work are diverse. In a series of studies [4–12], leprosy neuropathy is predominantly sensitive, heat and pain sensitivities are the most compromised and, in general, it has an asymmetric pattern (multiple mononeuropathy).

If peripheral nerve changes are not identified, monitored and treated appropriately, the result is irreversible nerve damage, which can result in permanent deformity and disability [13]. Semmes-Weinstein monofilaments (SWM) are used to assess and monitor tactile sensation in specific territories of the nerve trunks of the hands and feet. The standard esthesiometer kit recommended by the Ministry of Health of Brazil is composed of six nylon monofilaments, 38 mm long and with different diameters, which exert a specific force that corresponds to weight variation from 0.07 to 300 gram-force (gf) [14].

Considering that the involvement of peripheral nerves is present in all clinical forms of leprosy, usually as an asymmetric peripheral neuropathy, predominantly sensitive [15], it is important to assess the patterns of hands and feet SWM-test at diagnosis, as well as their clinical-therapeutic follow-up in an objective way.

Our objectives were to evaluate patterns of SWM-test changes in the hands and feet of leprosy patients at diagnosis and their modifications at the end of treatment; to compare the number of SWM-test points classified as altered for hands and feet before and after treatment; to evaluate the frequency of alteration of the radial, ulnar and median nerves in the hands, and medial plantar, lateral plantar, sural and calcaneal branches of the tibial nerve in the feet by the points tested on SWM-test at diagnosis and at the end of treatment; to correlate the patterns of SWM-test alteration in the diagnosis with the responses to the Leprosy Suspicion Questionnaire (LSQ), as published before [16, 17]; to correlate the patterns of SWM-test changes for hands and feet at diagnosis with anti-PGL-I serology; to correlate the patterns of SWM-test changes for hands and feet at diagnosis and at the end of treatment with the degree of functional disability of the diagnosis and discharge; to correlate the patterns of SWM-test alteration (stable, improvement, worsening) for hands and feet at diagnosis and their clinical-therapeutic evolution at the end.

## Subjects and methods

### Type of study

This study is a longitudinal, prospective cohort study.

### Diagnostic criteria for leprosy

The subjects underwent a standardized clinical dermatoneurological exam according to Brazilian Ministry of Health guidelines as described in previous article from our group. [16, 17]. We classified the patients considering the guidelines adapted by Madrid (Congress of Madrid 1953) [18] and the Indian Association of Leprology (IAL 1982) [19] classifications, as follows: indeterminate (I), polar tuberculoid (T), borderline (B), polar lepromatous (L) and pure neural leprosy (PNL); and broadly classified according to WHO operational criteria [PB (I and T) and MB (B and L)] [16, 20].

### Ethics, consent and permissions

This study was approved by the Research Ethics Committee at the Clinical Hospital of Ribeirão Preto Medical School, University of São Paulo (protocol number 2.165.032, MH-Brazil). Written informed consent was obtained from every participant, including the parent/guardian of each participant younger than 18 years old. All procedures involving human subjects comply with the ethical standards of Declaration of Helsinki (1975/2008).

### Study sample

The SWM-test was performed by a single dermatologist and leprologist at the beginning and at the end of the treatment of patients diagnosed with leprosy in the municipality of Jardinópolis during the period from 2016 to 2019.

### Tactile sensitive test by Semmes-Weinstein Monofilaments (SWM-test)

The SW monofilament kit, consisting of six colored monofilaments, was used. Each color corresponds to a sensation threshold: green (0.07 gf), blue (0.2 gf), violet (2.0 gf), red (4.0 gf), orange (10.0 gf), pink (300 gf).

Considering the skin pressure threshold for one-point static touch was estimated with the SWM. The critical force is the axial force necessary to cause the filament to buckle. Initially, the test with the monofilaments was demonstrated to the patient in an arm area with normal sensation. After this stage, the test began with the SW monofilaments. With the patient's eyes closed, each monofilament was applied perpendicularly for about 1 to 2 seconds at each skin point inside the respective nerve sensitive territory (dermatome). The pressure on the skin should be applied for 1 to 2 seconds until the filament curvature is reached, not allowing it to slide over the skin [21, 22]. To assess the sensation in the path of the radial, ulnar and median nerves in the hands, and tibial, sural and saphenous nerves in the feet, the monofilament was applied in order to bend over the area without, however, sliding over the patient's skin. Seven points were evaluated on each hand (Fig 1) and 9 points on each foot (Fig 2). As recommended, the test was initiated by the thinnest monofilament and, therefore, having the lowest pressure (0.07 gf, green monofilament). When there was no response, the blue monofilament (0.2 gf) was used, and so on. The record was made on a simplified neurological assessment form, containing the color of the first filament perceived by the patient.

About unit of measurement, the use of the logarithmic Manufacturer´s filament Number (MN) was shown at least as early as 1978 [23] to be a source of confusion (these authors also pointed out that an engineering approach would require stress rather than pressure as the

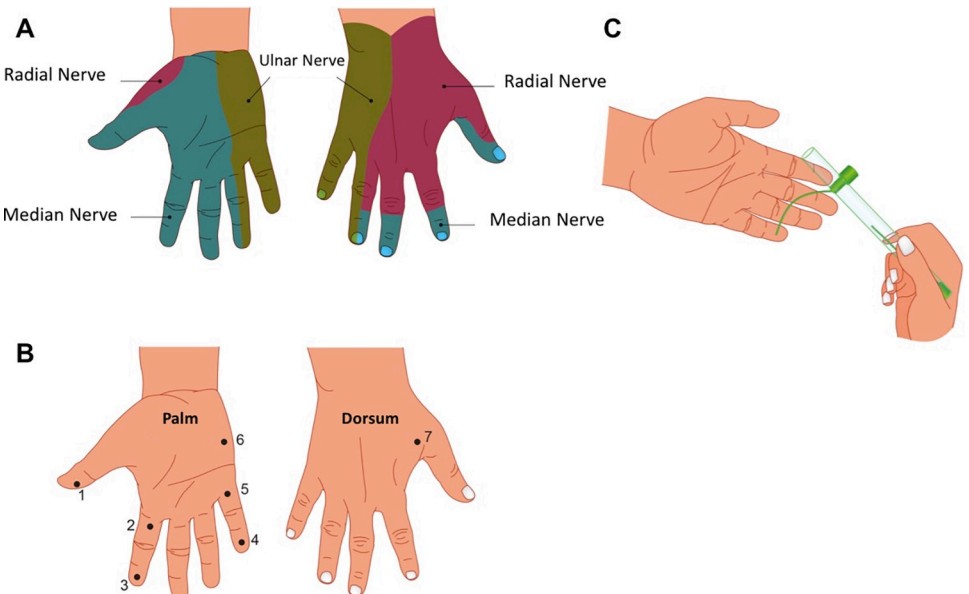

**Fig 1. Dermatomes of the radial, ulnar and median nerves and the 7-SWM-test points tested on the hand.** The normal tactile sensation threshold for the hand corresponds to the green monofilament (0.07 gf).

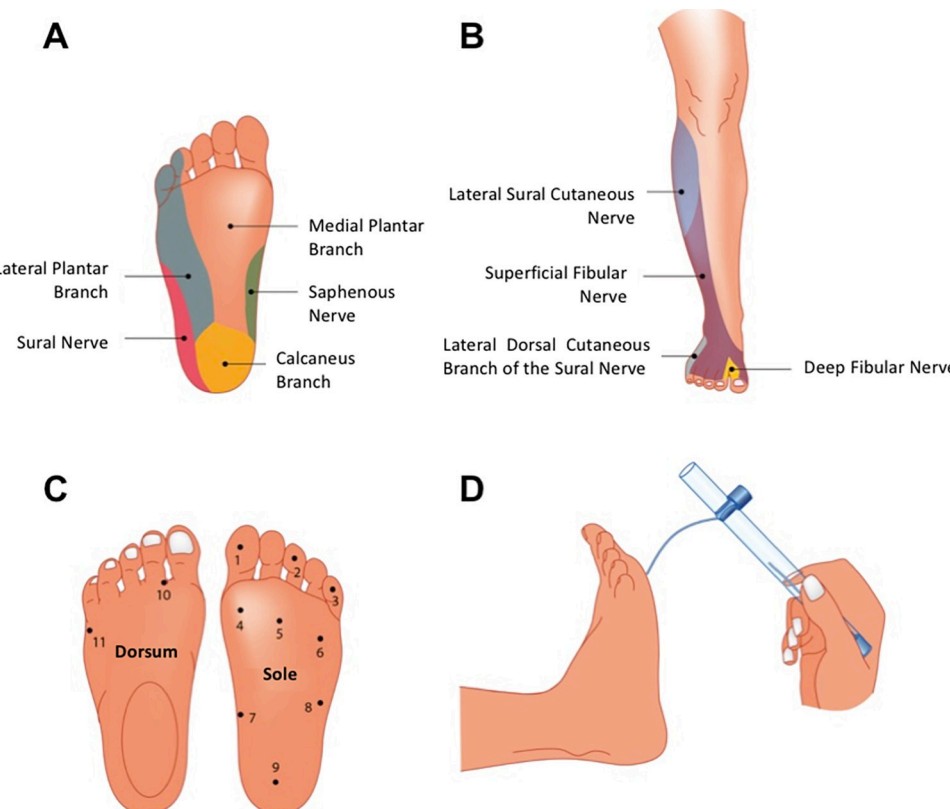

**Fig 2. Dermatomes of the branches of the fibular and tibial nerves and the 11-SWM-test points tested on the foot.** The normal tactile sensation threshold for the foot corresponds to the green and blue monofilaments (0.2 gf).

relevant variable) and was only introduced (by Weinstein) to facilitate the graphic visualization and statistical treatment of results. Bell-Krotoski and Tomancik [24] have long recommended the simple use of the standard monofilament force values as they have repeatedly been shown to provide consistent and useful results for clinical evaluations and follow-ups over time. Thus, we considered the standard monofilament force values (gram-force) for all analyses in this study.

## Analysis and interpretation of SWM-test points

The respective points were plotted on an Excel spreadsheet considering the values according to the color of its sensation threshold defined by the SW-monofilament Kit: green (0.07 gf), blue (0.2 gf), violet (2.0 gf), red (4.0 gf), orange (10.0 gf), pink (300 gf). For the black one (when the pink monofilament is not recognized), the value of 400 gf was established for calculation purposes.

For the hands, the sensation to green monofilament (0.07 gf) was considered normal, while for the feet, it was up to the blue one (0.2 gf).

In the spreadsheet, each patient (rows) had its values described point-to-point (columns) in the respective hands and feet at diagnosis and at the end of treatment.

After the distribution of the points, using the Excel "ACCOUNT" tool and applying the criterion ">0.07" for the hand points and ">0.2" for the feet, the total numbers of altered SWM-test points for each hand and foot of each individual, at diagnosis and at the end of treatment, were obtained. From this, we calculated the percentage, the sum, the average, and the maximum value of points altered for each body part.

The Excel "ACCOUNT" tool was used in the analysis of the sample, for each point, applying criteria equal to the respective grammage value of each monofilament, such as "0.07", "0.2", "2", up to "400". Subsequently, their sum was calculated for the elaboration of graphs (bars and corresponding colors), both for absolute values and percentages, and for pre and post-treatment analysis.

For the analysis of the percentage of altered nerves in the hands and feet at diagnosis and at the end of treatment, the points of the dermatomes of the median (points 1, 2 and 3), ulnar (points 4, 5 and 6) and radial (point 7) nerves were considered for the hands; and the medial plantar branch (points 1, 2, 4, 5, 7), lateral plantar branch (points 3, 6), sural nerve (point 8) and calcaneal branch (point 9), for the feet. With the Excel "ACCOUNT" tool and applying the criterion ">0.07" for the hand points and ">0.2" for the foot ones, the number of altered points corresponding to each aforementioned nerve was obtained.

To analyze the therapeutic evolution of the sample point-by-point, only patients who completed the treatment were considered. All the respective right and left points were placed in a single column. From these values, initially trying to separate the altered points from the normal ones for hands and feet, the difference between the initial and final values was calculated. Its result showed us whether the evolution was "stable" ($\Delta = 0$), if there was "improvement" ($\Delta > 0$) or if it was "worse" ($\Delta < 0$). Thus, we calculated the percentage of evolution of the points and the respective average percentage of stability, improvement and worsening of all points.

Finally, considering that not all individuals improve their sensation by SWM-test to the point of being considered normal, it becomes important to know how satisfactory or not their evolution was in the analysis, for instance, there is an evolution from 300g, at the beginning, to 4g, at the end of the treatment. Therefore, a legend of points was established for the values of the respective differences between the initial and final assessments, ranging from stable, equal to zero, to +6 (improvement) or to -6 (worsening), as shown in Table 1.

In the distribution of each point, we calculated the absolute number of points that had the respective evolution (+6 to -6) and their respective percentages within the total point sample.

**Table 1. Standardization of SWM-test points and values given to the evolutionary differences between them.**

| Points | Dif. | Point value | Dif. | Point value | Dif. | Point value | Dif. | Point value | Dif. | Point value | Dif. | Point value |
|---|---|---|---|---|---|---|---|---|---|---|---|---|
| 400 | | | | | | | | | | | | |
| 300 | 100 | 1 | | | | | | | | | | |
| 10 | 390 | 2 | 290 | 1 | | | | | | | | |
| 4 | 396 | 3 | 296 | 2 | 6 | 1 | | | | | | |
| 2 | 398 | 4 | 298 | 3 | 8 | 2 | 2 | 1 | | | | |
| 0.2 | 399.8 | 5 | 299.8 | 4 | 9.8 | 3 | 3.8 | 2 | 1.8 | 1 | | |
| 0.07 | 399.95 | 6 | 299.95 | 5 | 9.95 | 4 | 3.96 | 3 | 1.98 | 2 | 0.15 | 1 |

Dif. difference

### Assessment of anti-PGL-I titer by ELISA

Indirect ELISA was used to measure the anti-PGL-I IgM titer of every serum sample using the protocol previously reported [16, 17, 25]. The sample index was calculated by dividing their optical density (O.D.) per the established 0.295 cut-off; indexes above 1.0 were considered positive.

### Patients' follow-up

The follow-up of patients diagnosed with leprosy in the municipality of Jardinópolis from 2016 to 2019 was carried out by the dermatologist responsible for the study. The following variables were considered to assess stability, improvement or worsening of dermatological signs: hypochromatic macules with alteration of sensation and/or some dysautonomia, localized irregular patches of circumscribed hair loss and appearance of ichthyosis. In relation to neurological symptoms, the following were considered: electric shock-like pain on nerve palpation, tingling, cramps, numbness and needle sensation.

### Statistical analysis

Test-T for paired samples was used to assess asymmetry for each sensation point tested on the right and on the left. The correlation between the sum of the number of altered SWM-test points per individual in the diagnosis and the sum of the number of responses to the LSQ, the values of the anti-PGL-I antibody indexes and the degree of functional disability were analyzed. Finally, the correlation between the sum of the number of altered SWM-test points per individual and the degree of functional disability at the end of the treatment was evaluated. The Binomial Logistic Regression Analysis was performed in order to assess the association age, sex and number of pre-treatment altered SWM-test points with the outcome of having a SWM-test clinical improvement using the jamovi project (2021). *jamovi* (Version 1.6) [Computer Software]. Retrieved from https://www.jamovi.org.

### Results

The demographic characterizations at diagnosis and at the end of treatment, as well as the clinical classification of patients are described in Table 2. All patients were multibacillary, 81 (75.7%) patients had disability at diagnosis, while 34 (45.3%) had some disability at the end of treatment.

Regarding SWM-test, it was performed in 107 patients at diagnosis and in 76 patients at the end of treatment. The reasons for not evaluating patients at the end of treatment are described

**Table 2. Demographic characterization and clinical classification of patients.**

|  | Diagnose (n = 107) | | End of treatment (n = 76) | |
|---|---|---|---|---|
| Sex | N | % | n | % |
| Male | 48 | 44.9 | 34 | 44.7 |
| Female | 59 | 55.1 | 42 | 55.3 |
| Age (years) | | | | |
| Min | 6 | | 7 | |
| Max | 77 | | 78 | |
| Average | 42.3 | | 42.8 | |
| Median | 45 | | 45.5 | |
| Age range | | | | |
| < 15 | 10 | 9.3 | 7 | 9.2 |
| 15 \|— 20 | 6 | 5.6 | 6 | 7.9 |
| 20 \|— 30 | 13 | 12.1 | 7 | 9.2 |
| 30 \|— 40 | 17 | 15.9 | 12 | 15.8 |
| 40 \|— 50 | 16 | 15.0 | 11 | 14.5 |
| 50 \|— 60 | 26 | 24.3 | 17 | 22.4 |
| 60 \|— 70 | 12 | 11.2 | 11 | 14.5 |
| ≥ 70 | 7 | 6.5 | 5 | 6.6 |
| Classification | | | | |
| Borderline | 101 | 94.4 | 70 | 92.1 |
| Lepromatous | 1 | 0.9 | 1 | 1.3 |
| Pure neural leprosy | 5 | 4.7 | 5 | 6.6 |
| Degree of disability | | | | |
| 0 | 26 | 24.3 | 42 | 55.3 |
| 1 | 63 | 58.9 | 21 | 27.6 |
| 2 | 18 | 16.8 | 13 | 17.1 |
| Not evaluated | 0 | - | 31 | - |

in Table 3. At diagnosis, 61 patients (57%) had normal SWM-test in the hands and only 13 (12.1%) in the feet. At the end of treatment, 57 (75%) patients had no abnormal point in the hands and 31 (40.8%) in the feet, an improvement differential of 18% for the hands, and 28.7% for the feet.

Out of the 76 patients evaluated at the end of treatment, in 48 (63.2%) of them the number of abnormal SWM-test points to the hands remained stable compared to the initial. On the other hand, there was improvement in 22 (28.9%) of them and worsening in only 6 (7.9%) patients. Regarding the assessment of the feet tactile sensation, the number of abnormal

**Table 3. Reasons for not performing SWM-test in 31 patients.**

| Reasons | n | % |
|---|---|---|
| Completed treatment, but did not return for end-of-treatment evaluation | 3 | 9.7 |
| Patients who moved out of town | 6 | 19.4 |
| Leprosy treatment dropout | 7 | 22.6 |
| Refused diagnosis and treatment | 1 | 3.2 |
| Still under treatment | 12 | 38.7 |
| Died before the end of treatment | 2 | 6.5 |

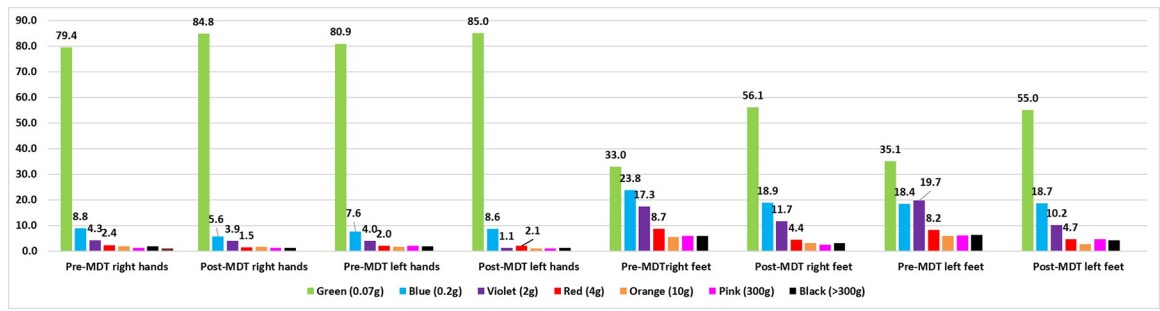

**Fig 3. Side-by-side distribution of the percentage of the number of SWM-test points of the hands and feet at diagnosis and after the end of treatment of patients diagnosed with leprosy in Jardinópolis from 2016 to 2019.**

SWM-test points remained stable in 17 (22.4%) patients, while there was an improvement in 47 (61.8%) and worsening in 12 (15.8%) patients.

The improvement in patients' SWM-test can be assessed by increasing the percentage of points considered normal for the hands (green point = 0.07 gram-force) and feet (green and blue points = 0.2 gf), as well as the decrease in percentages referring to the other points from violet (0.2 gf) to black (300 gf is not recognized) as represented in Figs 3 and 4. The most significant increase was that of green points for the feet at the end of treatment.

Statistical differences were observed in the evolution of the means of the summation of the number of altered points which had been detected by SWM-test performed on the right and left hands and feet of each patient when compared at diagnosis and at the end of treatment with WHO multidrug therapy, except for the left hand (p> 0.07), as shown in Fig 5.

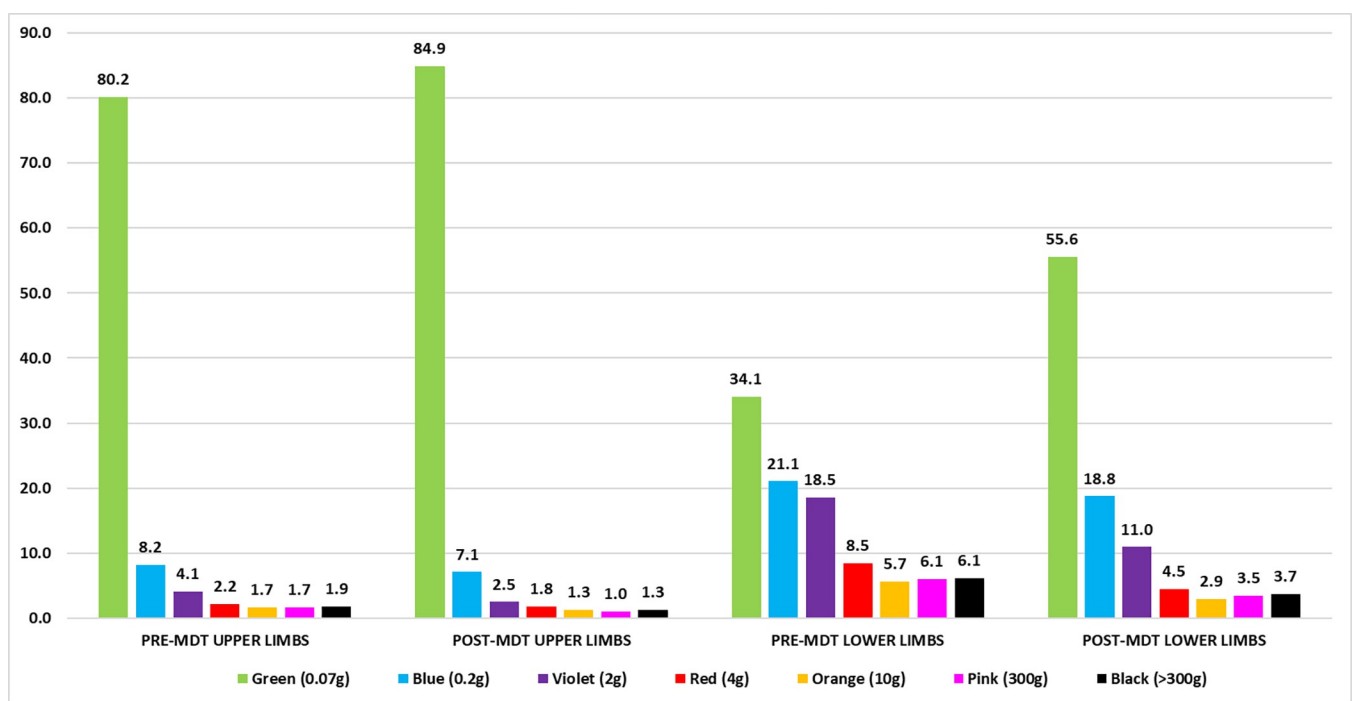

**Fig 4. Percentage distribution of the number of SWM-test points of the upper and lower limbs at diagnosis and after the end of treatment of patients diagnosed with leprosy in Jardinópolis from 2016 to 2019.**

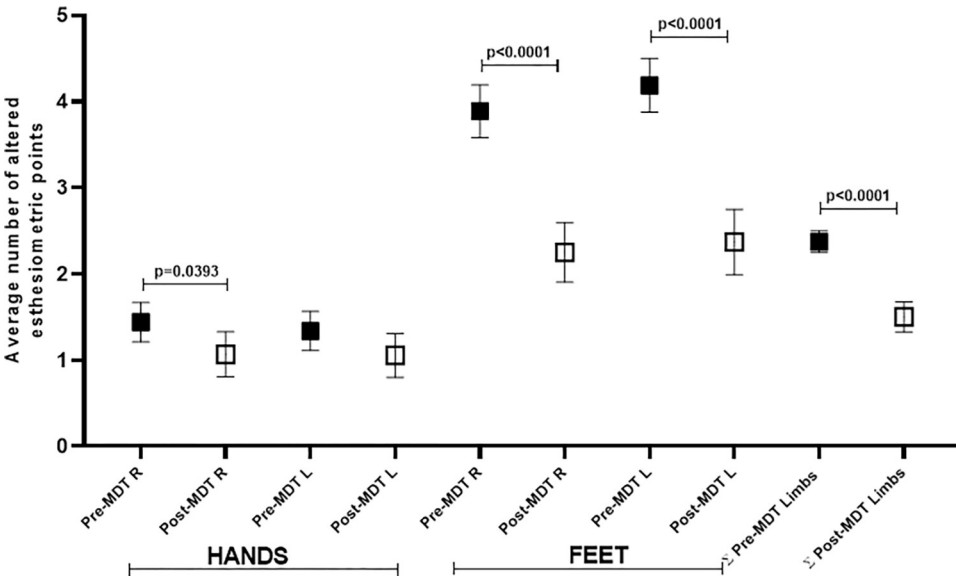

**Fig 5. Evolution of the averages of the sum of the number of altered points detected by SWM-test performed on the right and left hands and feet of each patient when compared at diagnosis and at the end of WHO MDT.**

Considering the sum of the number of altered SWM-test points distributed in the hands and feet, respectively, for each patient, significant differences were observed in their therapeutic evolution, as shown in Fig 6.

We seek to characterize the point-to-point evolution as stable, worsening (negative values) or improvement (positive values), as described in Table 4, by taking into account that each

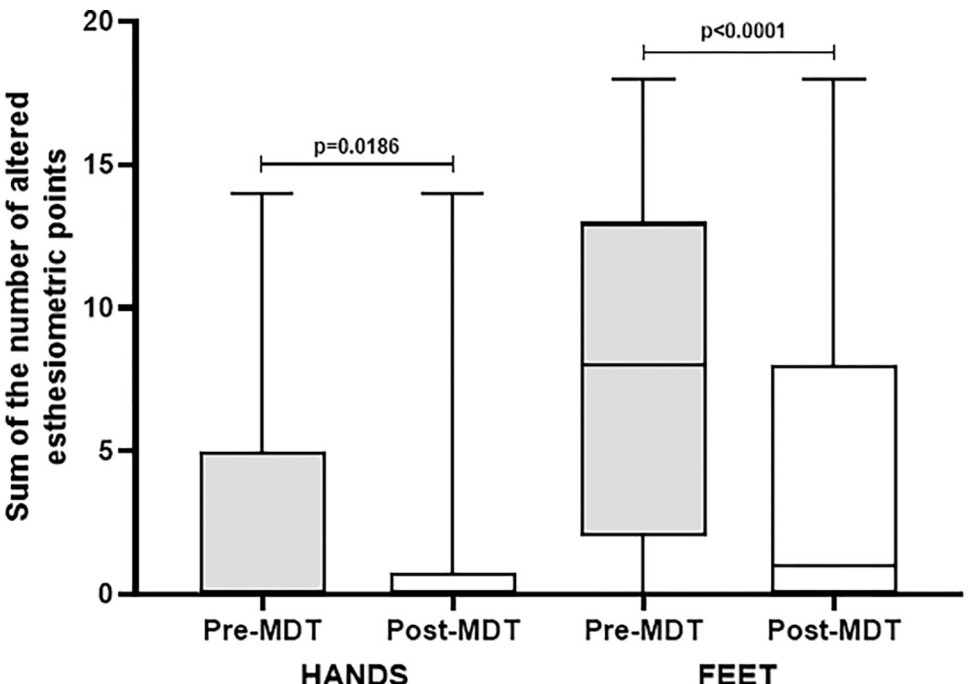

**Fig 6. Evolution of the sum of the number of altered SWM-test points in the hands and feet of leprosy patients evaluated at diagnosis and at the end of WHO multidrug therapy in Jardinópolis (SP).**

**Table 4. Variation of the point-to-point evolution by SWM-test of the hands and feet at the end of the leprosy patient treatment in Jardinópolis (SP).**

| Variation / evolution | SWM-TEST POINTS OF THE HANDS | | | | | | | SWM-TEST POINTS OF THE FEET | | | | | | | | | TOTAL | % |
|---|---|---|---|---|---|---|---|---|---|---|---|---|---|---|---|---|---|---|
| | PT1 | PT2 | PT3 | PT4 | PT5 | PT6 | PT7 | PT1 | PT2 | PT3 | PT4 | PT5 | PT6 | PT7 | PT8 | PT9 | | |
| -6 | 0 | 0 | 0 | 0 | 0 | 0 | 0 | 0 | 0 | 0 | 0 | 0 | 0 | 0 | 0 | 0 | 0 | 0.0 |
| -5 | 0 | 0 | 0 | 0 | 0 | 0 | 0 | 0 | 0 | 0 | 1 | 0 | 1 | 0 | 0 | 0 | 2 | 0.1 |
| -4 | 0 | 1 | 1 | 0 | 0 | 0 | 0 | 0 | 0 | 1 | 0 | 0 | 1 | 1 | 0 | 0 | 5 | 0.2 |
| -3 | 1 | 2 | 0 | 2 | 0 | 0 | 0 | 1 | 1 | 1 | 2 | 1 | 2 | 1 | 1 | 1 | 16 | 0.7 |
| -2 | 2 | 0 | 0 | 0 | 1 | 1 | 0 | 3 | 1 | 1 | 2 | 4 | 3 | 1 | 2 | 2 | 23 | 0.9 |
| -1 | 3 | 4 | 6 | 5 | 7 | 9 | 9 | 9 | 7 | 8 | 14 | 9 | 13 | 11 | 20 | 11 | 145 | 6.0 |
| STABLE | 125 | 132 | 135 | 132 | 127 | 122 | 122 | 82 | 96 | 83 | 72 | 76 | 62 | 94 | 64 | 45 | 1569 | 64.6 |
| 1 | 11 | 11 | 7 | 7 | 9 | 8 | 15 | 34 | 28 | 40 | 32 | 32 | 38 | 30 | 40 | 40 | 382 | 15.7 |
| 2 | 7 | 0 | 1 | 0 | 0 | 3 | 3 | 14 | 14 | 11 | 16 | 19 | 21 | 7 | 21 | 25 | 162 | 6.7 |
| 3 | 0 | 1 | 0 | 2 | 5 | 4 | 1 | 6 | 0 | 3 | 7 | 2 | 5 | 1 | 1 | 23 | 61 | 2.5 |
| 4 | 2 | 0 | 2 | 2 | 1 | 2 | 2 | 1 | 1 | 1 | 3 | 3 | 5 | 5 | 1 | 2 | 33 | 1.4 |
| 5 | 0 | 0 | 0 | 0 | 0 | 2 | 0 | 1 | 1 | 1 | 1 | 2 | 0 | 0 | 1 | 3 | 12 | 0.5 |
| 6 | 0 | 1 | 0 | 2 | 2 | 1 | 0 | 1 | 2 | 1 | 2 | 2 | 1 | 1 | 0 | 0 | 16 | 0.7 |

point could evolve to improvement or worsening, regardless of whether it was characterized as altered or not, that is, a point recognized as 300 gf at diagnosis that became 4 gf at the end of the treatment indicates a significant improvement in sensation, remaining, however, as an altered SWM-test point since it is greater than 0.07 gf for the hands and 0.2 gf for the feet.

In this point-to-point analysis, it was observed that there was an improvement of 27.5% of the points (666), and in 24.9%, the variation was from +1 to +3, while the worsening occurred in only 7.9% of the points (191), with a variation of -1 in 6%, and stability occurred in 64.6% of the points (1569).

The serological result of the anti-PGL-I antibody was positive in 38 (40%) patients, 16 (42.1%) with index ≥2, as shown in Table 5.

In Spearman's correlation analysis, a positive correlation with statistical significance was observed between the number of hand and foot SWM-test altered points at diagnosis and the degree of disability at diagnosis (0.69) and between the number of hand and foot SWM-test altered points and the degree of disability at the end of treatment (0.80). The correlation between the number of hand and foot SWM-test altered points at diagnosis and the Elisa anti-PGL-I index was low (0.027).

The number and percentage of individuals, sum, average and maximum number of SWM-test altered points in the hands and feet at diagnosis and at the end of treatment are described in Table 6.

The distribution of the number of altered SWM-test points with the corresponding nerves for hands and feet, before and after treatment, are described in Tables 7–10.

**Table 5. Results of anti-PGL-I antibody measurements (anti-PGL-I index; *cut off* 0.295).**

| | Total (n = 95) | |
|---|---|---|
| | n | % |
| Anti-PGL-I < 1 (negative) | 57 | 60.0 |
| Anti-PGL-I ≥ 1 (positive) | 38 | 40.0 |
| 1.0 |— 1.5 | 14 | 36.8 |
| 1.5 |—2.0 | 8 | 21.1 |
| ≥ 2.0 | 16 | 42.1 |

**Table 6. The number and percentage of individuals with altered points in the hands and feet by SWM-test at diagnosis and at the end of treatment, and the sum, average and maximum number of altered points in the hands and feet at diagnosis and at the end of treatment.**

|  | Evaluation at diagnosis | | | | Evaluation at the end of treatment | | | |
|  | Hands | | Feet | | Hands | | Feet | |
|  | R | L | R | L | R | L | R | L |
|---|---|---|---|---|---|---|---|---|
| n of patients with altered SWM-test points | 38 | 38 | 90 | 88 | 18 | 17 | 41 | 38 |
| % of patients with altered SWM-test points | 35.5 | 35.5 | 84.1 | 82.2 | 23.7 | 22.4 | 53.9 | 50.0 |
| Sum of the number of altered SWM-test points | 154 | 143 | 416 | 448 | 81.0 | 80.0 | 171 | 180 |
| Average of altered SWM-test points/patient | 4.1 | 3.8 | 4.6 | 5.1 | 4.5 | 4.7 | 4.2 | 4.7 |
| Maximum number of altered SWM-test points/patient | 7 | 7.0 | 9.0 | 9.0 | 7.0 | 7.0 | 9.0 | 9.0 |

n: number; %: percentage; R: right; L: left.

At diagnosis, among all the points evaluated by the respective corresponding nerves considering both sides, the median nerve (3 points each side) presented 115/642 altered SWM-test points (17.9%), the ulnar (3 points each side) presented 135/642 (21.03%) altered SWM-test points, while the radial (single point) presented 47/214 (21.96%) altered SWM-test points. Bilateral alterations were found in 35 (30.4%) median nerves, 47 (34.8%) ulnar nerves and 18 (38.3%) radial nerves. Considering the values obtained by SWM-test for each point, in the upper limbs, corroborating the higher percentage of altered SWM-test points, only point 7 (sensitive radial nerve) showed significant asymmetry (p = 0.04), while among the median (p = 0.15) and the ulnar (p = 0.12) points there was no demonstrated asymmetry.

Additionally, considering the sum of the number of altered SWM-test points in the upper limbs (hands), there was no statistical difference (p = 0.18) between the right and left sides.

At the end of treatment, considering the 76 patients, the median nerve showed 38/456 altered SWM-test points (14.9%), the ulnar nerve showed 56/456 (12.3%) altered SWM-test points, while the radial nerve presented 24/152 (16.9%) altered SWM-test points. Bilateral alterations were found in 35 (30.4%) median nerves, 47 (34.8%) ulnar nerves and 18 (38.3%) radial nerves. Considering the values obtained by SWM-test for each point, in the upper limbs, despite a higher percentage of altered points, point 7 no longer showed significant asymmetry (p = 0.16), like the ulnar nerve (p = 0.12). The median nerves, however, became asymmetrical after treatment (p = 0.009).

In the lower, limbs among all the evaluated points that correspond to the tibial nerve considering both sides, 864 of the 1926 points tested (44.9%) were reported as altered (> 0.2 gf), demonstrating a significant asymmetry (p = 0.005) at diagnosis.

**Table 7. Distribution of the number of SWM-test (normal and altered) points of the hands per patient at treatment diagnosis and corresponding nerves.**

|  | Median nerve (Points 1, 2, 3) | | | | Ulnar nerve (Points 4, 5, 6) | | | | Radial nerve (Point 7) | | | |
|  | R | | L | | R | | L | | R | | L | |
|  | N | % | n | % | n | % | n | % | n | % | n | % |
|---|---|---|---|---|---|---|---|---|---|---|---|---|
| No altered point | 78 | 72.9 | 83 | 77.6 | 76 | 71.0 | 79 | 73.8 | 85 | 79.4 | 82 | 76.6 |
| 1 altered point | 9 | 8.4 | 10 | 9.3 | 9 | 8.4 | 7 | 6.5 | 22 | 20.6 | 25 | 23.4 |
| 2 altered points | 6 | 5.6 | 0 | 0 | 6 | 5.6 | 4 | 3.7 | - | - | - | - |
| 3 altered points | 14 | 13.1 | 14 | 13.1 | 16 | 15.0 | 17 | 15.9 | - | - | - | - |
| Total | 107 | 100 | 107 | 100 | 107 | 100 | 107 | 100 | 107 | 100 | 107 | 100 |

n: number; %: percentage; R: right; L: left.

**Table 8. Distribution of the number of SWM-test (normal and altered) points of the hands per patient at the end of the treatment and corresponding nerves.**

| | Median nerve (Points: 1, 2, 3) | | | | Ulnar nerve (Points 4, 5, 6) | | | | Radial nerve (Point 7) | | | |
|---|---|---|---|---|---|---|---|---|---|---|---|---|
| | R | | L | | R | | L | | R | | L | |
| | n | % | n | % | n | % | n | % | n | % | n | % |
| No altered point | 61 | 80.3 | 64 | 84.2 | 61 | 80.3 | 63 | 82.9 | 65 | 85.5 | 63 | 82.9 |
| 1 altered point | 3 | 3.9 | 2 | 2.6 | 5 | 6.6 | 0 | 0 | 11 | 14.5 | 13 | 17.1 |
| 2 altered points | 1 | 2.3 | 2 | 2.6 | 3 | 3.9 | 2 | 2.6 | - | - | - | - |
| 3 altered points | 11 | 14.5 | 8 | 10.5 | 7 | 9.2 | 11 | 14.5 | - | - | - | - |
| Total | 76 | 100 | 76 | 100 | 76 | 100 | 76 | 100 | 76 | 100 | 76 | 100 |

n: number; %: percentage; R: right; L: left.

Separating them by the corresponding branches, the medial plantar branch (5 points per side) presented alteration in 436/1070 points (40.7%), the lateral plantar branch in 162/428 points (37.9%), the sural nerve in 111/214 points (51.9%) and the calcaneal branch in 172/214 points (80.4%).

Bilateral alterations were found in 95/436 (21.8%) of the medial plantar branches, 51/162 (31.5%) of the lateral plantar branches, 41/111 (36.9%) of the sural nerves and 80/172 (46.5%) of the calcaneal branches. Considering the values obtained by SWM-test for each point in the lower limbs, the asymmetries within the individual values were statistically significant between the medial plantar branches ($p = 0.03$) and within the sural nerves ($p = 0.003$). Nevertheless, within the points of the lateral plantar ($p = 0.38$) and calcaneal ($p = 0.1$) branches, there was no demonstrated asymmetry.

Additionally, due to the sum of the number of altered SWM-test points in the lower limbs (feet), the difference also demonstrated significant asymmetry ($p = 0.04$), which highlights the importance of the tibial nerve for the establishment of asymmetric leprosy neuropathy.

As for the lower limbs (feet), at the end of treatment, there was an important reduction in the percentage of alterations in the medial plantar branch with 167/760 altered SWM-test points (21.9%), the lateral plantar branch with 66/304 points (21/7%), the sural nerve with 43/152 points (28.3%) and the calcaneal branch with 75/152 points (49.3%).

Considering the values obtained by SWM-test for each point in the lower limbs, the asymmetry within the individual values remained statistically significant in the medial plantar

**Table 9. Distribution of the number of SWM-test (normal and altered) points of the feet per patient at the treatment diagnosis and corresponding nerves.**

| | Medial plantar branch (Points 1, 2, 4, 5, 7) | | | | Lateral plantar branch (Points 3, 6) | | | | Sural nerve (Point 8) | | | | Calcaneal branch (Point 9) | | | |
|---|---|---|---|---|---|---|---|---|---|---|---|---|---|---|---|---|
| | R | | L | | R | | L | | R | | L | | R | | L | |
| | n | % | n | % | n | % | N | % | n | % | n | % | n | % | n | % |
| No altered point | 44 | 41.1 | 41 | 38.3 | 54 | 50.5 | 44 | 41.1 | 49 | 45.8 | 54 | 50.5 | 21 | 19.6 | 21 | 19.6 |
| 1 altered point | 14 | 13.1 | 10 | 9.3 | 24 | 22.4 | 26 | 24.3 | 58 | 54.2 | 53 | 49.5 | 86 | 80.4 | 86 | 80.4 |
| 2 altered points | 13 | 12.1 | 16 | 15.0 | 29 | 27.1 | 37 | 34.6 | - | - | - | - | - | - | - | - |
| 3 altered points | 11 | 10.3 | 12 | 11.2 | - | - | - | - | - | - | - | - | - | - | - | - |
| 4 altered points | 8 | 7.5 | 9 | 8.4 | - | - | - | - | - | - | - | - | - | - | - | - |
| 5 altered points | 17 | 15.9 | 19 | 17.8 | - | - | - | - | - | - | - | - | - | - | - | - |
| Total | 107 | 100 | 107 | 100 | 107 | 100 | 107 | 100 | 107 | 100 | 107 | 100 | 107 | 100 | 107 | 100 |

n: number; %: percentage; R: right; L: left.

**Table 10. Distribution of the number of SWM-test (normal and altered) points of the feet per patient at the end of the treatment and corresponding nerves.**

| | Medial plantar branch (Points 1, 2, 4, 5, 7) | | | | Lateral plantar branch (Points 3, 6) | | | | Sural nerve (Point 8) | | | | Calcaneal branch (Point 9) | | | |
|---|---|---|---|---|---|---|---|---|---|---|---|---|---|---|---|---|
| | R | | L | | R | | L | | R | | L | | R | | L | |
| | n | % | n | % | n | % | n | % | n | % | n | % | N | % | n | % |
| No altered point | 51 | 67.1 | 50 | 65.8 | 56 | 73.7 | 55 | 72.4 | 52 | 68.4 | 57 | 75.0 | 37 | 48.7 | 40 | 52.6 |
| 1 altered point | 4 | 5.3 | 5 | 6.6 | 9 | 11.8 | 7 | 9.2 | 24 | 31.6 | 19 | 25.0 | 39 | 51.3 | 36 | 47.4 |
| 2 altered points | 6 | 7.9 | 2 | 2.6 | 11 | 14.5 | 14 | 18.4 | - | - | - | - | - | - | - | - |
| 3 altered points | 6 | 7.9 | 3 | 3.9 | - | - | - | - | - | - | - | - | - | - | - | - |
| 4 altered points | 2 | 2.6 | 8 | 10.5 | - | - | - | - | - | - | - | - | - | - | - | - |
| 5 altered points | 7 | 9.2 | 8 | 10.5 | - | - | - | - | - | - | - | - | - | - | - | - |
| Total | 76 | 100 | 76 | 100 | 76 | 100 | 76 | 100 | 76 | 100 | 76 | 100 | 76 | 100 | 76 | 100 |

n: number; %: percentage; R: right; L: left.

branches (p = 0.02). They became asymmetrical within the points of the lateral plantar branches (p = 0.2) and symmetrical in the sural branches (p = 0.13), but they remained symmetrical in the calcaneal branches (p = 0.28).

As for the number of SWM-test points with coinciding bilateral changes, despite a decrease in the number of altered points with the treatment, there was an increase in the percentages, with 61/167 (36.5%) points of the medial plantar branches, 25/66 (37.9%) of the lateral plantar branches, 18/43 (55.8%) of the sural nerves and 36/75 (48%) of the calcaneal branches.

Considering all the points tested in the feet as being from the tibial nerves, a reduction was observed in 351 of the 1368 points tested (25.7%) at the end of treatment. Nevertheless, the asymmetry between the sides remained significant (p = 0.001).

In summary, the distribution of patients by the number of affected limbs at diagnosis defined by SWM-test and the respective averages of the number of altered SWM-test points per limb are shown in Table 11.

Only 11/107 (10.2%) individuals did not present alterations to SWM-test at diagnosis, among which the clinical and laboratory criteria stood out: hypochromatic macules with alterations in sensation (n = 10), erythematous plaque with alteration in sensation (n = 1), LSQ positivity (n = 9), incomplete endogenous histamine test (n = 6), and positive anti-PGL-I ELISA index (n = 6) with a mean value of 1.4.

**Table 11. Distribution of leprosy patients by number of affected limbs as defined by SWM-test and the respective averages of the sum of the number of altered points per limb and their evolution.**

| n of individuals | n of affected limbs/individual at the diagnose | Average of the sum of the number of altered SWM-test points per limb | | | | | | | |
|---|---|---|---|---|---|---|---|---|---|
| | | n diagnose | | | | n end of treatment | | | |
| | | RUL | LUL | RLL | LLL | RUL | LUL | RLL | LLL |
| 11 | 0 | 0 | 0 | 0 | 0 | 0.0 | 0.0 | 0.1 | 0.1 |
| 12 | 1 | 0.5 | 0.1 | 0.7 | 0.4 | 0.0 | 0.0 | 0.5 | 0.1 |
| 40 | 2 | 0.0 | 0.0 | 3.4 | 4.0 | 0.2 | 0.2 | 1.2 | 1.4 |
| 14 | 3 | 1.2 | 1.4 | 4.8 | 4.9 | 0.0 | 0.1 | 0.7 | 1.0 |
| 30 | 4 | 4.4 | 4.1 | 6.9 | 7.2 | 3.4 | 3.3 | 5.0 | 5.1 |

n number; RUL right upper limb; LUL left upper limb; RLL right lower limb; LLL left lower limb.

The logistic regression model was significant ($X^2 = 39.0$; p<0.001; $R^2$ MacFadden = 0.41; Accuracy = 0.82; Specificity = 0.47; Sensitivity = 0.98; AUC = 0.84), demonstrating an absence of association between age [OR = 1.01 (%CI = 0.97–1.06); p = 0.5], sex [OR = 1.1 (%CI = 0.3–4.4); p = 0.8], number of pre-treatment altered SWM-test points [OR = 0.9 (%CI = 0.8–1.03); p = 0.2] with the clinical outcome (SWM-test clinical improvement). Statistical report is in S1 File.

## Discussion

The study mapped alterations in the sensation of the hands and feet by the SWM-test of 107 patients at diagnosis and in 76 patients at the end of treatment for leprosy patients followed in Jardinópolis from 2016 to 2019. The study highlighted alterations in the patterns of hand and foot SWM-test at diagnosis and their consequent modifications to the specific treatment of leprosy with antimicrobial drugs, which defines the infectious (bacterial) etiology of neuropathy and reinforces the importance of using SWM-test in the clinical diagnosis of the disease.

Assessing sensation is especially important for preventing peripheral neuropathy. Semmes-Weinstein monofilaments are widely used in the assessment of tactile sensation in peripheral neuropathies of the upper and lower limbs [14, 22]. Although there is other device to assess the sensation for diagnosis and the follow-up of the neuropathy as the pressure-specified sensory device (PSSD), described by Baltodano et al [26], with perhaps a marginal improvement over the SWM, the major advantages in assessing the sensation of peripheral nerves by using Semmes-Weinstein monofilaments are: its easy application, low cost and the fact that the patient is blindfolded, making it difficult to have false responses in the test [27].

Since it is an infection marked especially by the involvement of peripheral nerves, leprosy causes damage related to sensory, motor, and autonomic functions. Sensitive impairment, present in all forms of the disease, often precedes the involvement of motor function [15]. The best approach for the prevention of neural damage and the consequences caused by leprosy is early diagnosis and appropriate treatment. However, certain patients will still need actions to prevent the progression of the disability, or even rehabilitation measures [6, 28]. This group may consist of the following cases: new cases already detected with some disability; cases that will develop some type of disability during or after the end of treatment, and old cases diagnosed late with disability already installed [29].

Regarding demographic characteristics, the number of female patients was higher than that of male patients, in contrast to the official Brazilian data 2014–2018, with higher rates among men than women in all age groups. The distribution of patients by age group followed the same patterns of the Brazilian data [30]. The high number of diagnoses in children under 15 years of age (9.3%), all patients being multibacillary, the high number of patients with disability at diagnosis and the high seroprevalence of anti-PGL-I are evidence of hidden endemicity revealed in the Jardinópolis municipality during the research period, as already demonstrated in the literature [16, 20].

Hand and foot SWM-test at the end of treatment was not performed in 31 (29.0%) patients. Despite having completed the 12-month treatment, three patients did not return for the discharge assessment. These patients were actively searched, but they were not found. Regarding the 6 patients who moved away from Jardinópolis, a medical report was made available, in addition to immediate communication from the State's epidemiological surveillance network. Despite the presence of a dermatologist/leprologist in patient care, there was a high number (22.6%) of treatment dropout. Leprosy treatment dropout occurs due to several factors, among them: lack of knowledge about the disease, turnover of health professionals, reaction episodes, worsening of neurological symptoms, especially in the first months of treatment, side effects of

the treatment and the lack of credibility in leprosy cure [31]. The patient who refused the diagnosis and treatment had multiple hypo-anesthetic hypochromatic and anhydrotic macules with histamine areflexia, besides positive PCR for *Mycobacterium leprae* in slit skin smear. At the departure of the dermatologist from the municipality, in November 2019, 12 patients were still undergoing leprosy treatment. The deaths of 2 patients were due to complications of chronic obstructive pulmonary disease in a smoking patient and stroke in a hypertensive and diabetic patient.

Cutaneous innervation of the hands is done by the median, ulnar and radial nerves. The innervation of the dorsum of the foot and the lateral face of the leg is performed by the lateral sural cutaneous, superficial fibular, lateral dorsal cutaneous and deep fibular branches of the common fibular nerve. The cutaneous innervation of the plantar region is performed by branches of the tibial nerve, such as: calcaneal, medial plantar, lateral plantar and saphenous [32]. SWM-test of the hands and feet allows the mapping of tactile sensation thresholds in the path of the radial, ulnar and median nerves in the hands, and tibial and common fibular nerves in the feet, providing a record that can be easily compared and interpreted, both in diagnosis and in its therapeutic evolution. In the upper limbs, it is important to highlight the importance of point 7 (radial nerve), which was significantly asymmetrical. In the lower limbs (feet), it is worth noting that the difference between the sum of altered SWM-test points showed significant asymmetry (p = 0.04) when comparing the sides, which highlights the importance of the tibial nerve for the establishment of asymmetric leprosy neuropathy. In addition, the asymmetry defined in point 8 exclusively reinforces the focus of the aggression, which associated with the asymmetry of SWM-test alterations are characteristic of leprosy neuropathy.

The greater involvement of the nerves of the lower limbs, represented by the higher frequency of altered SWM-test points on the feet, both before and after treatment, differs from the previous literature, which suggests that the nerves of the upper limbs are affected more frequently than the lower ones [12, 33]. Several studies demonstrate that among the nerves that showed the highest frequency of involvement, assessed by electroneuromyography, there are: the ulnar, the superficial radial, the sural, the superficial fibular, the sensitive tibial; and those which presented the lowest frequencies are: the common fibular nerve and the median [4–12].

The data on the prevalence of neural impairment in leprosy are controversial and vary according to the region, the instruments of detection and the evaluation criteria. It is worth mentioning that only the points of the plantar region that are correlated to the dermatomes of the tibial nerve branches were tested. Such nerve is not usually addressed in the conventional electroneuromyography exam, and, undoubtedly, it becomes the most affected nerve in leprosy.

The points on the dorsum of the foot, point 10 (deep fibular nerve) and point 11 (sural nerve), which are relative to the dermatomes of the branches of the common fibular nerve, were not tested. Thus, it is estimated that the involvement of the nerves of the lower limbs is even greater.

All patients were treated with a standard multibacillary multidrug (MDT/WHO) regimen, consisting of the respective antimicrobial drugs and doses, namely: monthly supervised dose with 600 mg rifampicin + 100 mg dapsone + 300 mg clofazimine, and self-administered daily dose with 100 mg dapsone + clofazimine 50 mg. Early diagnosis and treatment prevents permanent loss of neural function. *Mycobacterium leprae* has tropism for the peripheral nerves, binding to Schwann cells, in such a way that it induces demyelination [30]. Therefore, the improvement in neurological symptoms and patients' SWM-test, represented by the increase in the percentage of points considered normal for the hands (green point = 0.07 g-f) and feet (green and blue points = 0.2 g-f), as well as the decrease in the percentages referring to the other points from violet (2 g-f) to black (it does not recognize 300 g-f), with antibiotic

treatment is strong evidence of the infectious etiology of peripheral neuropathy and it is very valuable in the follow-up of leprosy patients, especially in those with laboratory tests (SSS and skin biopsy) without the identification of *M. leprae*.

These findings are corroborated by the significant 31% reduction in the degree of functional disability after treatment, in addition to the 26% average reduction in the number of altered SWM-test points in the feet, with emphasis on the calcaneal branch of the tibial nerve, while an average reduction of 7.5% of the points of the hands specially in the ulnar nerve. Moreover, in the point-to-point analysis, the SWM-test evolution at the end of the treatment showed an improvement in 27.5% of all the points analyzed, varying from 1 to 3 levels of weight in 25% of these points.

Although the Leprosy Suspicion Questionnaire has proved to be an important tool in the leprosy diagnosis, essentially due to the neurological complaints of numbness, tingling, nerve pain, among others [16, 17], there was no correlation between the sum of the number of questions to the LSQ and the number of altered points to the hand and foot SWM-test, which denotes the multiple and multifocal character of leprosy neuropathy.

Considering the distribution of the sample by limb involvement and the respective averages of the sum of the number of altered SWM-test points, it is worth highlighting that only 10.2% of the patients did not present changes to the SWM-test at diagnosis; however other clinical and laboratory criteria had already defined diagnosis, which demonstrates the importance of this tool for the definition of neuropathy and leprosy diagnosis in approximately 90% of cases.

In addition, the reduction in the values of these averages after the end of multidrug therapy demonstrates to what extent the treatment was able to modify the sensation profile, especially of the feet of these patients, and the most marked improvement being was evident among patients with fewer affected limbs, reaffirming the proposition that early diagnosis and treatment are essential in preventing deformities and disability. It is worth emphasizing that those patients who, even after multidrug therapy, still have residual disability detected by the SWM test, may benefit from appropriate peripheral nerve decompression surgery, preventing definitive neural damage from leprosy, as described by Wan et al [34, 35]. Furthermore, considering the binomial logistic regression analysis, we emphasize that regardless of the initial severity weight based on the number of altered SWM-test points, the treatment was undoubtedly able to recover the sensation of hands and feet.

## Conclusion

The tactile sensation test using Semmes-Weisntein monofilaments proved to be an important instrument for defining peripheral neuropathy of leprosy patients in the study, affecting 89.7% (96) of them and reaching 1 to 4 limbs. 19.8% of the altered SWM-test points in the hands, and 44.8% in the feet were detectable, evolving to 15.1% in the hands and to 25.6% in the feet at the end of the treatment. Both variations had statistical significance.

At diagnosis, using the hand and foot SWM-test criteria, the tibial was the most affected nerve by leprosy, mainly in the calcaneal (80.4%) and sural branches (52%). However, the asymmetries were significant regarding the values of the medial plantar (points 1, 2, 4, 5, and 7) and the sural (points 3 and 6) branches. In the hands, the nerve with the highest absolute number of affected points was the ulnar, even though, in percentage, the radial nerve was not only more affected, but also the only definer of asymmetry.

There was neither satisfactory correlation between the patterns of SWM-test alteration at diagnosis and the responses to the LSQ nor in relation to the results of anti-PGL-I serology. Nevertheless, there was a significant positive correlation with the degree of functional disability at diagnosis (0.69) and at the end of treatment (0.80).

SWM-test in point-to-point analysis showed improvement in 27.5% of the points with anti-microbial treatment ranging from 1 to 3 points in graduation, worsening in only 7.9% of the points and stability in 64.9% of the points.

Finally, hand and foot tactile sensation test using Semmes-Weinstein monofilaments proved to be effective in establishing the asymmetric and focal sensitive neuropathic pattern, constituting an important complementary exam in leprosy diagnosis, besides reaffirming its role in the documentation and proof of the therapeutic efficacy of MDT, a low-cost and easy-to-implement instrument which should be an indispensable tool for health care professionals, both in primary and specialized care in leprosy.

## Supporting information

**S1 File. Statistical report.**
(PDF)

**S1 Data.**
(XLSX)

## Author Contributions

**Conceptualization:** Marco Andrey Cipriani Frade, Fred Bernardes Filho.

**Data curation:** Marco Andrey Cipriani Frade, Fred Bernardes Filho.

**Formal analysis:** Marco Andrey Cipriani Frade, Fred Bernardes Filho.

**Funding acquisition:** Marco Andrey Cipriani Frade.

**Investigation:** Marco Andrey Cipriani Frade, Fred Bernardes Filho, Claudia Maria Lincoln Silva, Glauber Voltan, Filipe Rocha Lima, Thania Loyola Cordeiro Abi-Rached, Natália Aparecida de Paula.

**Methodology:** Marco Andrey Cipriani Frade, Fred Bernardes Filho, Natália Aparecida de Paula.

**Project administration:** Marco Andrey Cipriani Frade, Fred Bernardes Filho.

**Resources:** Marco Andrey Cipriani Frade, Fred Bernardes Filho.

**Writing – original draft:** Marco Andrey Cipriani Frade, Fred Bernardes Filho.

**Writing – review & editing:** Marco Andrey Cipriani Frade, Fred Bernardes Filho, Natália Aparecida de Paula.

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
