## [Decision Letter · Decision Letter 0]

3 Aug 2021

PONE-D-21-17910

Semmes-Weinstein Monofilament esthesiometry for the definition of patterns of sensitivity alteration of the hands and feet in leprosy diagnosis and monitoring

PLOS ONE

Dear Dr. Frade,

Thank you for submitting your manuscript to PLOS ONE. After careful consideration, we feel that it has merit but does not fully meet PLOS ONE’s publication criteria as it currently stands. Therefore, we invite you to submit a revised version of the manuscript that addresses the points raised during the review process.

We look forward to receiving your revised manuscript.

Kind regards,

Fabio Santanelli, di Pompeo d'Illasi, MD, PhD

Academic Editor

PLOS ONE

Additional Editor Comments (if provided):

Reviewers' comments:

Reviewer's Responses to Questions

**Comments to the Author**

1. Is the manuscript technically sound, and do the data support the conclusions?

Reviewer #1: Yes

Reviewer #2: Yes

2. Has the statistical analysis been performed appropriately and rigorously? 

Reviewer #1: Yes

Reviewer #2: I Don't Know

3. Have the authors made all data underlying the findings in their manuscript fully available?

Reviewer #1: Yes

Reviewer #2: Yes

4. Is the manuscript presented in an intelligible fashion and written in standard English?

Reviewer #1: Yes

Reviewer #2: No

5. Review Comments to the Author

Reviewer #1: Congratulation to the authors for their study “Semmes-Weinstein Monofilament esthesiometry for the definition of patterns of sensitivity alteration of the hands and feet in leprosy diagnosis and monitoring”. Overall the manuscript is insightful and contributes to the subject of nervous impairment in leprosy patients, specifically MB. The study is very detailed and meticulous. It addresses some gaps found in literature. However, we do have some concerns worth addressing:

1. You state repeatedly throughout the manuscript that you only included multibacillary (or “lepromatous”) cases of leprosy in the study. However, Table 2 details how 101 patients from the initial population were actually classified as Borderline. Therefore, I must inquire:

- Which diagnostic criteria did you use to define the patients? The WHO classification, The Ridley-Jopling classification? (PMID: 17366457) Please define in Materials and Methods, for the sake of clarity.

- Why did you decide not to include patients with paucibacillary leprosy in this study? This should also be stated in the manuscript for the sake of transparency. Perhaps include it as one of the limits of the study.

2. You stress the importance of tibial involvement in leprosy, which is not something new to date. You should consider better implementing what is already known in literature and discuss it in the manuscript (PMID: 4795534, which is also specific for MB patients, and PMID: 11391190).

3. You analyse the esthesiometric improvement in the tested nerves after treatment with WHO regimen. However, not all included patients started with the same severity. In fact, you should consider implementing a multivariate statistical analysis to understand the weight of severity on sensory recovery after treatment. Perhaps also consider evaluating the effects of age and gender as well.

4. You should better discuss the limitations in the study’s methodology. Specifically, you should at least mention alternatives that have been deemed superior in esthesiometric assessments, even if for different purposes, such as the PSSD (PMID: 26220428, PMID: 32128706).

5. The way serology and anti-PDL-1 positivity affected esthesiometric changes is unclear. Although it is true that some evidence suggests that it could be a useful tool for evaluating the efficacy of treatment (i.e. higher serology levels in untreated patients) [PMID: 19784481], one must wonder how relevant is this section for the purpose of your study? Especially considering that correlation index was found to be low.

6. Some minor critiques to the content of the manuscript should also be addressed:

• Consider implementing the type of study (i.e. cohort study) in the title of the manuscript;

• Introduction is not engaging enough, too long and too focused on academic concepts which add very little to the manuscript. Consider reworking it in order to make it more focused on the aims and endpoints of the study;

• Consider moving the following sections of the manuscript to the Discussion: “In a series of studies […] fibular nerve and the median” (p. 4, lines 75-80);

• Consider moving the following sections of the manuscript to the Materials and Methods: “The standard esthesiometer kit […] the dermatomes of each nerve” (p. 4-5, lines 84-92); “All patients were treated […] + clofazimine 50 mg” (p. 26, lines 474-477).

• Finally, some very minor language concerns: “focallity” instead of focality (p.3, line 44); “surreal” instead of sural (p. 22, line 375). Please do consider using a native English speaker for a quick grammar and spelling check.

Reviewer #2: 1)there is very little that is new in in this manuscript. The SWM(S"deemmes-Weinstein monofilament)and ball point pen and the PSSD (pressure-specified sensory device) have been reported almost endlessly to show that medical management is effective. This study does confirm these classic findings , and yet there is no reference to this, espeially not even one mention of the PSSD, which has been shown to be more sensitive in identifying early cases of Leprosy. At the end, I will include a few references that must be added

2) I am sorry to point out all that the SWM estimates the sensibility of a point of skin. it does not make a true measurement. As this paper proves, a set of 6 filaments were used, and therefore there are no normative data but an estimate of a range. Did each filament chosen as the correct one for that point of skin have the number of the filament,, for example 2.83, first converted from its logarithmic form before statistics That number on the filament is the force (not pressure) of that piece of skin and is the log to the base 10 of that force in 0.1mg. So if pressure is going to be discussed, each measurement of force must be changed to pressure by dividing by the diameter of the filament used.

3) The SWM measures sensibility of a piece of skin and yet the authors refer to these as dermatomes. Clearly they are measuring a piece of the skin territory of a peripheral nerve, and not a dermatome. See Figures 1 and 2 for example as well as the text.

4) rather paper should say skin pressure threshold for one point static touch was estimated with the SWM.

when the above is cleared up, and tables, corrected, and numerical pressure not force values are used, I am happy to reconsider this paper

here are recent articles showing what these authors show with medical treatment, but these apply to nerve decompression, which is the disability these patients have after medical treatment (also not mentioned in this paper)

Baltodano, PA, Wan, E, Noboa, J, Rosson, GD, Dellon, AL, Selecting a Test for

Leprous Neuropathy Screening, J Reconstr Microsurg, 31:607-613, 2015.

Wan, E, Rivadeniera, AF, Serrano, HA, Napit, I, Garbino, JA, Joshua, J, Cardona-Castro, N,

Dellon, AL, Theuvenet, W, Protocol for a randomized controlled trial investigating

decompression for leprous neuropathy (the DELN protocol), Lepr Rev, 87:553-561,2016.

Wan, EL, Noboa, J., Baltodono, PA, Jousin, RM, Erickson, W, Wilton, JP, Rosson, GD,

Dellon, AL, Nerve decompression for leprous neuropathy: A prospective study from

Ecuador, Lepr Review, 88:95-108, 2017.

6. PLOS authors have the option to publish the peer review history of their article (what does this mean?). If published, this will include your full peer review and any attached files.

Reviewer #1: **Yes: **Gennaro D'orsi

Reviewer #2: No

---

## [Author Response · Author response to Decision Letter 0]

25 Aug 2021

August 25, 2021.

Fabio Santanelli, di Pompeo d'Illasi, MD, PhD

Academic Editor

PLOS ONE

Sub: PONE-D-21-17910

Dear Editors,

We would like to thank you for all comments about our manuscript: “Semmes-Weinstein Monofilament esthesiometry for the definition of patterns of sensitivity alteration of the hands and feet in leprosy diagnosis and monitoring” because they appointed some important aspects to improve it.

The requested edits were considered and they are in a red in the manuscript. At the end of this authors’ responses is the statistical report. We also included the statistical report as a supporting information file.

Yes, I received support for my study from all of the following sources:

-WHO Implementation Research Team of Ribeirão Preto Medical School

-Center of National Reference in Sanitary Dermatology focusing on Leprosy of Ribeirão Preto Clinical Hospital

-Brazilian Health Ministry (MS/FAEPAFMRP-USP)

-Fiocruz Ribeirão Preto - TED 163/2019

Yes, you can update my Financial Disclosure statement as follows on my behalf:

"This study was supported by the WHO Implementation Research Team of Ribeirão Preto Medical School in the form of a grant awarded to MACF (771/2016 SCAPIR), the Center of National Reference in Sanitary Dermatology focusing on Leprosy of Ribeirão Preto Clinical Hospital, Ribeirão Preto, São Paulo, Brazil in the form of funds awarded to MACF, the Brazilian Health Ministry (MS/FAEPAFMRP-USP) in the form of grants awarded to MACF (749145/2010, 767202/2011), and Fiocruz Ribeirão Preto - TED 163/2019 in the form of a grant awarded to MACF (Processo: N° 25380.102201/2019-62/ Projeto Fiotec: PRES-009-FIO-20)."

Comments to the Author

1. Is the manuscript technically sound, and do the data support the conclusions?

Reviewer #1: Yes

Reviewer #2: Yes

2. Has the statistical analysis been performed appropriately and rigorously?

Reviewer #1: Yes

Reviewer #2: I Don't Know

3. Have the authors made all data underlying the findings in their manuscript fully available?

Reviewer #1: Yes

Reviewer #2: Yes

4. Is the manuscript presented in an intelligible fashion and written in standard English?

Reviewer #1: Yes

Reviewer #2: No

5. Review Comments to the Author

Reviewer #1: Congratulation to the authors for their study “Semmes-Weinstein Monofilament esthesiometry for the definition of patterns of sensitivity alteration of the hands and feet in leprosy diagnosis and monitoring”. Overall the manuscript is insightful and contributes to the subject of nervous impairment in leprosy patients, specifically MB. The study is very detailed and meticulous. It addresses some gaps found in literature. However, we do have some concerns worth addressing:

1. You state repeatedly throughout the manuscript that you only included multibacillary (or “lepromatous”) cases of leprosy in the study. However, Table 2 details how 101 patients from the initial population were actually classified as Borderline. Therefore, I must inquire:

- Which diagnostic criteria did you use to define the patients? The WHO classification, The Ridley-Jopling classification? (PMID: 17366457) Please define in Materials and Methods, for the sake of clarity.

Author’s comments: Sure, you are right. We did not consider describe about it in the manuscript. In lines 102 to 109, we have added: 

Diagnostic criteria for leprosy

The subjects underwent a standardized clinical dermatoneurological exam according to Brazilian Ministry of Health guidelines as described in previous article from our group. [16,17]. We classified the patients considering the guidelines adapted by Madrid (Congress of Madrid 1953) [18] and the Indian Association of Leprology (IAL 1982) [19] classifications, as follows: indeterminate (I), polar tuberculoid (T), borderline (B), polar lepromatous (L) and pure neural leprosy (PNL); and broadly classified according to WHO operational criteria [PB (I and T) and MB (B and L)] [16,20].

- Why did you decide not to include patients with paucibacillary leprosy in this study? This should also be stated in the manuscript for the sake of transparency. Perhaps include it as one of the limits of the study.

Author’s comments: Thanks for your comment. Following the classification criteria based on both cutaneous and nerve evaluations by the specialist, the clinical classification of the patients as paucibacillary become difficult because most of the patients presented more than one nerve impairment added to skin signs of leprosy.

2. You stress the importance of tibial involvement in leprosy, which is not something new to date. You should consider better implementing what is already known in literature and discuss it in the manuscript (PMID: 4795534, which is also specific for MB patients, and PMID: 11391190).

Author’s comments: Thanks for your comment. We highlighted the importance of the tibial involvement in leprosy patients as compared to other nerves mainly to upper limbs. Both articles that you mentioned described interesting studies about tibial nerve exclusively different of our goal to show the importance of esthesiometric tool for the timely diagnosis of leprosy.

About the papers, the first paper by Swift et al (1973), they studied only about the motor conduction velocity, and they did not study the sensory nerves. On the other hand, on the paper from Richard et al (2001), they selected nine leprosy patients with exclusive tibial nerve palsy causing a loss of protective sensation on the sole of the foot, and they considered a very late loss of sensation defined only when they not feeling a 10-g monofilament in at least three sites on the foot.

3. You analyse the esthesiometric improvement in the tested nerves after treatment with WHO regimen. However, not all included patients started with the same severity. In fact, you should consider implementing a multivariate statistical analysis to understand the weight of severity on sensory recovery after treatment. Perhaps also consider evaluating the effects of age and gender as well.

Author’s comments: We thank you for your considerations to improve our paper. We have made the multivariate statistical analysis using a binomial logistic regression analysis, and to get a good performance of this statistical model we considered as independent variables age, sex, number of altered esthesiometric points in pre-treatment, the score of sensory recovery (0 to 5 according each five recovered points), and finally as outcome the esthesiometric clinical improvement (dependent variable) which it was established by the difference between the summation of the number of pre-treatment altered points and the post-treatment ones becoming to nominal variable.

The logistic regression model was significant (X2=39.0; p<0.001; R2 MacFadden=0.41; Accuracy=0.82; Specificity=0.47; Sensitivity=0.98; AUC=0.84), demonstrating an absence of association between age [OR=1.01 (%CI=0.97-1.06); p=0.5], sex [OR=1.1 (%CI=0.3-4.4); p=0.8], number of pre-treatment altered esthesiometric points [OR=0.9 (%CI=0.8-1.03); p=0.2] with the clinical outcome (esthesiometric clinical improvement). Therefore, independent of weight of severity of number of altered esthesiometric points in pre-treatment, after the treatment the sensory recovering happened.

In summary, we improved the manuscript adding:

1) Statistical analysis topic: lines 228-231: “The Binomial Logistic Regression Analysis was performed in order to assess the association age, sex and number of pre-treatment altered esthesiometric points with the outcome of having a esthesiometric clinical improvement using the jamovi project (2021). jamovi (Version 1.6) [Computer Software]. Retrieved from https://www.jamovi.org.”

2) Results topic: lines 410-414: “The logistic regression model was significant (X2=39.0; p<0.001; R2 MacFadden=0.41; Accuracy=0.82; Specificity=0.47; Sensitivity=0.98; AUC=0.84), demonstrating an absence of association between age [OR=1.01 (%CI=0.97-1.06); p=0.5], sex [OR=1.1 (%CI=0.3-4.4); p=0.8], number of pre-treatment altered esthesiometric points [OR=0.9 (%CI=0.8-1.03); p=0.2] with the clinical outcome (esthesiometric clinical improvement).”

3) In the end of the discussion topic: lines 533-535. “…Furthermore, considering the binomial logistic regression analysis, we emphasize that regardless of the initial severity weight based on the number of altered esthesiometric points, the treatment was undoubtedly able to recover the sensitivity of hands and feet.”

4. You should better discuss the limitations in the study’s methodology. Specifically, you should at least mention alternatives that have been deemed superior in esthesiometric assessments, even if for different purposes, such as the PSSD (PMID: 26220428, PMID: 32128706).

Author’s comments: Sure, you are right. We mentioned about this method to assess the sensitivity as PSSD as described by Baltodano et al (2015) in the second paragraph of discussion. (lines 426-429).

We have added the (red) test below:

“…Although there is other device to assess the sensitivity for diagnosis and the follow-up of the neuropathy as the PSSD (pressure-specified sensory device), described by Baltodano et al (2015), with perhaps a marginal improvement over the SWM, the major advantages using Semmes-Weinstein monofilaments are: its easy application, low cost and the fact that the patient is blindfolded, making it difficult to have false responses in the test [27].”

5. The way serology and anti-PDL-1 positivity affected esthesiometric changes is unclear. Although it is true that some evidence suggests that it could be a useful tool for evaluating the efficacy of treatment (i.e. higher serology levels in untreated patients) [PMID: 19784481], one must wonder how relevant is this section for the purpose of your study? Especially considering that correlation index was found to be low.

Author’s comments: Thanks for your comments. About this point of view, we highlighted that we found around 82% of patients (Table 6) with some altered esthesiometric point while the anti-PGL-I positivity was 40% only (Table 5).

6. Some minor critiques to the content of the manuscript should also be addressed:

• Consider implementing the type of study (i.e. cohort study) in the title of the manuscript;

Author’s comments: Thank you so much about this comment. Really, this new title will improve the quality and importance of our work. We changed the title to:

“Semmes-Weinstein Monofilament esthesiometry for the definition of patterns of sensitivity alteration of the hands and feet in a cohort leprosy study”

• Introduction is not engaging enough, too long and too focused on academic concepts which add very little to the manuscript. Consider reworking it in order to make it more focused on the aims and endpoints of the study;

Author’s comments: Thanks for your comment. We agreed with you. We cut the firs paragraph completely and changed a bit the second one. Additionally, considering the comment below, we moved the cited paragraph to the discussion part.

• Consider moving the following sections of the manuscript to the Discussion: “In a series of studies […] fibular nerve and the median” (p. 4, lines 75-80);

Author’s comments: Answered above.

• Consider moving the following sections of the manuscript to the Materials and Methods: “The standard esthesiometer kit […] the dermatomes of each nerve” (p. 4-5, lines 84-92); “All patients were treated […] + clofazimine 50 mg” (p. 26, lines 474-477).

Author’s comments: Thanks a lot. We have moved them for the methods.

• Finally, some very minor language concerns: “focallity” instead of focality (p.3, line 44); “surreal” instead of sural (p. 22, line 375). Please do consider using a native English speaker for a quick grammar and spelling check.

Author’s comments: Thanks a lot. We changed them.

Reviewer #2:

1)there is very little that is new in in this manuscript. The SWM(S"deemmes-Weinstein monofilament)and ball point pen and the PSSD (pressure-specified sensory device) have been reported almost endlessly to show that medical management is effective. This study does confirm these classic findings , and yet there is no reference to this, espeially not even one mention of the PSSD, which has been shown to be more sensitive in identifying early cases of Leprosy. At the end, I will include a few references that must be added

Author’s comments: Sure, thanks a lot for your comment. We mentioned about this method to assess the sensitivity as PSSD as described by Baltodano et al (2015) in the second paragraph of discussion. (lines 426-429). 

We have added the (red) test below: 

“…Although there is other device to assess the sensitivity for diagnosis and the follow-up of the neuropathy as the PSSD (pressure-specified sensory device), described by Baltodano et al (2015), with perhaps a marginal improvement over the SWM, the major advantages using Semmes-Weinstein monofilaments are: its easy application, low cost and the fact that the patient is blindfolded, making it difficult to have false responses in the test [27].”

2) I am sorry to point out all that the SWM estimates the sensibility of a point of skin. it does not make a true measurement. As this paper proves, a set of 6 filaments were used, and therefore there are no normative data but an estimate of a range. Did each filament chosen as the correct one for that point of skin have the number of the filament,, for example 2.83, first converted from its logarithmic form before statistics That number on the filament is the force (not pressure) of that piece of skin and is the log to the base 10 of that force in 0.1mg. So if pressure is going to be discussed, each measurement of force must be changed to pressure by dividing by the diameter of the filament used.

Author’s comments: Thanks a lot. We agree with you, but some points should be clarified according to the references below: 

The SWM test uses a subjective approach to evaluate the threshold of sensitivity of an area of skin: “does the patient perceive the application of a standard filament of specific critical force?” The critical force is the axial force necessary to cause the filament to buckle. The pressure over a nerve ending or group can be estimated, if necessary, for purposes of comparison, either by dividing the filament’s critical force by the area of the tip of the filament in contact with the skin during application, (possibly varying during application) or by the estimated area of skin which is effectively depressed over the relevant nerve endings during the application. Traditionally the calculation is simply Critical Force / Tip Area.

The use of the logarithmic Manufacturer´s filament Number (MN) was shown at least as early as 1978 (Levin S, et al.) to be a source of confusion (these authors also pointed out that an engineering approach would require Stress rather than Pressure as the relevant variable), and was only introduced (by Weinstein) to facilitate the graphic visualization and statistical treatment of results. It certainly does not help to convert the traditional MN numbers back to gram force values as they have long become irrelevant. For example, the widely used 10 gram-force filament, chosen by Birke and Sims in 1986 to identify the threshold between presence or absence of protective sensation in the plantar surface of the foot, would have to be renumbered “5.00” instead of the much-vaunted “5.07”, which is the log value for 11.749 grams-force. Incidentally, the filament cited in that paper as “75 grams” is quoted as having the MN 6.10, which would suggest that they may have been using a 125 gram filament.

On the other hand, Bell-Krotoski and others have long recommended the simple use of the standard monofilament force values as they have repeatedly been shown to provide consistent and useful results for clinical evaluations and follow-ups over time. The pressure applied or stress does not need to be imposed as if it were a more scientific measure.

LEVIN, S; PEARSALL, G; RUDERMAN, R J. Von Frey's method of measuring pressure sensibility in the hand: an engineering analysis of the Weinstein-Semmes pressure aesthesiometer. The Journal of hand surgery, v. 3, n. 3, p. 211-216, 1978.

BIRKE, J. A.; SIMS, D. S. Plantar sensory threshold in the ulcerative foot. Leprosy review, v. 57, n. 3, p. 261-267, 1986.

PFAU, D. B., HAROUN, O., LOCKWOOD, D. N., MAIER, C., SCHMITTER, M., VOLLERT, J., ... & TREEDE, R. D. Mechanical detection and pain thresholds: comparability of devices using stepped and ramped stimuli. Pain Reports, 5(6), 2020.

BELL-KROTOSKI, J., & TOMANCIK, E. The repeatability of testing with Semmes-Weinstein monofilaments. The Journal of hand surgery, 12(1), 155-161. 1987.

3) The SWM measures sensibility of a piece of skin and yet the authors refer to these as dermatomes. Clearly they are measuring a piece of the skin territory of a peripheral nerve, and not a dermatome. See Figures 1 and 2 for example as well as the text.

Author’s comments: Thanks for your comments. We changed the manuscript to become clear our message about the esthesiometry.

Lines 130-132: With the patient’s eyes closed, each monofilament was applied perpendicularly for Lines 151-157: about 1 to 2 seconds at each skin point inside the respective dermatome area.

We changed the title of the Figures 1 and 2.

4) rather paper should say skin pressure threshold for one point static touch was estimated with the SWM.

Author’s comments: Sure, we have added this message into the paragraph starting in line 127

“Considering the skin pressure threshold for one-point static touch was estimated with the SWM. Initially, the test with the monofilaments was demonstrated to the patient in an arm area with normal sensitivity. After this stage, the test began with the SW monofilaments. With the patient’s eyes closed, each monofilament was applied perpendicularly for about 1 to 2 seconds at each skin point inside the respective nerve sensitivity territory (dermatome).”

when the above is cleared up, and tables, corrected, and numerical pressure not force values are used, I am happy to reconsider this paper

Author’s comments: We understood your point of view, but about it, we clarified in Methods (lines 142-149) why we have used the standard monofilament force values as proposed by Bell-Krotoski et al (1987).

here are recent articles showing what these authors show with medical treatment, but these apply to nerve decompression, which is the disability these patients have after medical treatment (also not mentioned in this paper)

Baltodano, PA, Wan, E, Noboa, J, Rosson, GD, Dellon, AL, Selecting a Test for

Leprous Neuropathy Screening, J Reconstr Microsurg, 31:607-613, 2015.

Wan, E, Rivadeniera, AF, Serrano, HA, Napit, I, Garbino, JA, Joshua, J, Cardona-Castro, N,

Dellon, AL, Theuvenet, W, Protocol for a randomized controlled trial investigating

decompression for leprous neuropathy (the DELN protocol), Lepr Rev, 87:553-561,2016.

Wan, EL, Noboa, J., Baltodono, PA, Jousin, RM, Erickson, W, Wilton, JP, Rosson, GD,

Dellon, AL, Nerve decompression for leprous neuropathy: A prospective study from

Ecuador, Lepr Review, 88:95-108, 2017.

Author’s comments: Thanks a lot. We have cited Baltodano et al (2015) above and we included the last ones changing the last paragraph of discussion to:

“In addition, the reduction in the values of these averages after the end of multidrug therapy demonstrates to what extent the treatment was able to modify the sensitivity profile, especially of the feet of these patients, and the most marked improvement being was evident among patients with fewer affected limbs, reaffirming the proposition that early diagnosis and treatment are essential in preventing deformities and disability, avoiding the need for decompression surgery to prevent neural damage in leprosy as described by Wan, EL et al (2016 and 2017).”

Thank you for your consideration. I look forward to hearing from you.

Sincerely,

---

## [Decision Letter · Decision Letter 1]

13 Jan 2022

PONE-D-21-17910R1Semmes-Weinstein Monofilament esthesiometry for the definition of patterns of sensitivity alteration of the hands and feet in a cohort leprosy studyPLOS ONE

Dear Dr. Frade,

Thank you for submitting your manuscript to PLOS ONE. After careful consideration, we feel that it has merit but does not fully meet PLOS ONE’s publication criteria as it currently stands. Therefore, we invite you to submit a revised version of the manuscript that addresses the points raised during the review process.

We look forward to receiving your revised manuscript.

Kind regards,

Fabio Santanelli, di Pompeo d'Illasi, MD, PhD

Academic Editor

PLOS ONE

Reviewers' comments:

Reviewer's Responses to Questions

**Comments to the Author**

1. If the authors have adequately addressed your comments raised in a previous round of review and you feel that this manuscript is now acceptable for publication, you may indicate that here to bypass the “Comments to the Author” section, enter your conflict of interest statement in the “Confidential to Editor” section, and submit your "Accept" recommendation.

Reviewer #2: (No Response)

2. Is the manuscript technically sound, and do the data support the conclusions?

Reviewer #2: No

3. Has the statistical analysis been performed appropriately and rigorously? 

Reviewer #2: No

4. Have the authors made all data underlying the findings in their manuscript fully available?

Reviewer #2: Yes

5. Is the manuscript presented in an intelligible fashion and written in standard English?

Reviewer #2: No

6. Review Comments to the Author

Reviewer #2: SWM “esthesiometry” Leprosy Rev 2021

1. The title of the paper and throughout the paper misuses “esthesiometry” and “sensitivity” and these need to be changed throughout the paper.

The SWM estimate a range for one-point static touch sensibility, not sensitivity.

Esthesiometry is the measurement of tactile sensibility

The SWM cannot measure tactile sensibility and hence they cannot do esthesiometry,

even if this how the literature has misused these terms in the past. Time for perhaps

this paper to use the terms correctly.

Their title therefore might be: Evaluation of altered patterns of touch sensibility in

Leprosy using the Semmes-Weinstein monofilaments, and short title might me Patterns

of Sensibility in Hands and Feet of Leprosy Patients.

2. Abstract: This sensation is an example of how difficult it is to express what the authors are trying to say. They state “a 18% improvement in tactile sensitivity for the hands, and 28.7% for the feet was detected.” What can this possibly mean. Do they mean that 18% of their total number of hand patients had an improvement in sensation, or do they mean that the pre-treatment mean measurement has improved 18% for the hand.

3. Abstract: the next line is very confusing also: “In the hands, by esthesiometry, radial nerve was significantly asymmetric, while in the feet, the difference between the sum of altered esthesiometric points showed significant asymmetry when comparing both sides, highlighting the tibial nerve for the establishment of asymmetric leprosy neuropathy.” Can even the authors understand what this means????

4. The word “sensitivity” is still in this manuscript. Sensitivity is a statistical term as in sensitivity versus specificity. They use it wrongly when they should be saying sensibility.

5. Conclusion: Sadly ,the authors FAIL to discus that the remaining disability these patients have is due to chronic nerve compression, which can be helped by surgical decompression. They must emphasize that while the r value of .80 is great after triple drug therapy, r2 = 64 which means that 36% of meaning is unexplained or not improved. Those patients after triple drug therapy have residual disability that appropriate peripheral nerve surgery can help. References to this fact were added as refs 34 and 35 at the very last lines of the Discussion, however only to say that drug therapy can reduce the need for surgery, rather than adding the remaining patients would benefit from surgery, and this needs to be emphasized.

a) Wan, EL, Rivadeniera, AF, Jousin, RM, Dellon, AL, Treatment of Peripheral Neuropathy in Leprosy: The case for nerve decompression, Plast Reconstr Surg GO, 2016; 4:e637; doi10.1097/GO0000000000000641.

b) Wan, E, Rivadeniera, AF, Serrano, HA, Napit, I, Garbino, JA, Joshua, J, Cardona-Castro, N, Dellon, AL, Theuvenet, W, Protocol for a randomized controlled trial investigating decompression for leprous neuropathy (the DELN protocol), Lepr Rev, 87:553-561, 2016.

7. PLOS authors have the option to publish the peer review history of their article (what does this mean?). If published, this will include your full peer review and any attached files.

Reviewer #2: No

---

## [Author Response · Author response to Decision Letter 1]

24 Jan 2022

I put here specifically answers for the questions from the Reviewer #2 (also better writing in the attached document "response to reviewers")

1. The title of the paper and throughout the paper misuses “esthesiometry” and “sensitivity” and these need to be changed throughout the paper.

The SWM estimate a range for one-point static touch sensibility, not sensitivity.

Esthesiometry is the measurement of tactile sensibility. 

The SWM cannot measure tactile sensibility and hence they cannot do esthesiometry,

even if this how the literature has misused these terms in the past. Time for perhaps

this paper to use the terms correctly.

Their title therefore might be: Evaluation of altered patterns of touch sensibility in

Leprosy using the Semmes-Weinstein monofilaments, and short title might me Patterns

of Sensibility in Hands and Feet of Leprosy Patients.

Author’s comments: Thanks a lot, to advertise us about our writing and meanings. We have changed the title for: “Evaluation of altered patterns of tactile sensation in the diagnosis and monitoring of leprosy using the Semmes-Weinstein monofilaments” becoming shorter than before and maintained the main idea. Also, we changed the short title as your suggestion to: Hands and feet sensation patterns in leprosy. Therefore, as we have used in our works before in neuropathy field, we used the term “sensitivity” (the quality or condition of being sensitive), instead of “sensibility” (the ability to appreciate and respond to complex emotional or aesthetic influences). By the way, the term “sensation” is used very frequently also, and we decided to change for. On the other hand, we would like to maintain “diagnosis and monitoring” because we have used SWM for monitoring leprosy neuropathy during the leprosy treatment a long time, but our data demonstrated that the SWM also can help the health professionals to stablish the pattern of the leprosy neuropathy diagnosis. 

We have changed where we were using “esthesiometry” to SWM-test in all manuscript. Also, we have changed the using of the adjective “Esthesiometric” to SMW-test preceding the nouns throughout the paper marked in red writing. 

Initially, in lines 99-100, we excluded the sentence “also known as an esthesiometer” becoming: “Semmes-Weinstein monofilaments (SWM) are used to assess and monitor tactile sensation in specific territories of the nerve trunks of the hands and feet.”

In other important point, in Methods (page 6 – lines 153-154) we changed the subtitle Esthesiometry for “Tactile sensation test by Semmes-Weinstein Monofilaments (SWM-test)” to emphasize the correct meaning according to your suggestions (thanks again). 

2. Abstract: This sensation is an example of how difficult it is to express what the authors are trying to say. They state “a 18% improvement in tactile sensitivity for the hands, and 28.7% for the feet was detected.” What can this possibly mean. Do they mean that 18% of their total number of hand patients had an improvement in sensation, or do they mean that the pre-treatment mean measurement has improved 18% for the hand.

Author’s comments: Thanks for bring us your difficulty to understanding our test. We are sure they will become our work better. In the abstract we changed this part for: 

“At diagnosis, 81/107 (75.7%) patients had some degree of functional disability, and 46 (43%) of them had altered SWM-test in the hands and 94 (87.9%) of them in the feet. After one year of multibacillary multidrug therapy, the disability decreasing to 44/76 patients (57.9%) and decreasing of the percentual of altered SWM-test to 18% for the hands, and to 28.7% for the feet.”

Also, in the summary we have changed these considerations (marked in yellow): 

“At diagnosis, 43% of them had altered SWM-test in the hands, while 87.9% had it in the feet. After one year of multibacillary multidrug therapy, the percentual of patients with altered SWM-test decreased to 18% for the hands and 28.7% for the feet.

3. Abstract: the next line is very confusing also: “In the hands, by esthesiometry, radial nerve was significantly asymmetric, while in the feet, the difference between the sum of altered esthesiometric points showed significant asymmetry when comparing both sides, highlighting the tibial nerve for the establishment of asymmetric leprosy neuropathy.” Can even the authors understand what this means????

Author’s comments: Thanks for your comment. We changed the test to be clearer and more understandable. 

“In the hands, by SWM-test, only the radial nerve point demonstrated a significant asymmetry, while in the feet, the difference between the sum of altered SWM-test points showed significant asymmetry between both sides, highlighting the tibial nerve for the establishment of asymmetric leprosy neuropathy.”

4. The word “sensitivity” is still in this manuscript. Sensitivity is a statistical term as in sensitivity versus specificity. They use it wrongly when they should be saying sensibility.

Author’s comments: We understand your question, but we must disagree about your statement, as the term SENSITIVITY is not exclusive to statistics. According to several works in neuropathy, including some of our group, we have published it as SENSITIVITY, although some reviewers suggest that we change to "sensation", much more frequent than SENSIBILITY.

1. Frade MAC, Rosa DJF, Bernardes-Filho F, Spencer JS, Foss NT. Semmes-Weinstein monofila-ment: A tool to quantify skin sensation in macular lesions for leprosy diagnosis. Indian J Der-matol Venereol Leprol 2021; XX:1-9. (Added in our reference)

2. Fernanda Guzzo Gomes, Wilson Marques, Norma Tiraboschi Foss, Luísiane de Ávila Santana, Marco Andrey Cipriani Frade. Tactile threshold detection in leprosy patients with an electronic algometer. Journal of Neuroscience Methods, Volume 179, Issue 2, 2009, Pages 319-322,ISSN 0165-0270, https://doi.org/10.1016/j.jneumeth.2009.01.030.

3. Pedro J Tomaselli, Diogo F dos Santos, André C J dos Santos, Douglas E Antunes, Vanessa D Marques, Norma T Foss, Carolina L Moreira, Patrícia T B Nogueira, Osvaldo J M Nascimento, Luciano Neder, Amilton A Barreira, Marco A Frade, Isabela M B Goulart, Wilson Marques, Jr, Primary neural leprosy: clinical, neurophysiological and pathological presentation and pro-gression, Brain, 2021;, awab396, https://doi.org/10.1093/brain/awab396 (published in Octo-ber 2021)

5. Conclusion: Sadly ,the authors FAIL to discus that the remaining disability these patients have is due to chronic nerve compression, which can be helped by surgical decompression. They must emphasize that while the r value of .80 is great after triple drug therapy, r2 = 64 which means that 36% of meaning is unexplained or not improved. Those patients after triple drug therapy have residual disability that appropriate peripheral nerve surgery can help. References to this fact were added as refs 34 and 35 at the very last lines of the Discussion, however only to say that drug therapy can reduce the need for surgery, rather than adding the remaining patients would benefit from surgery, and this needs to be emphasized.

a) Wan, EL, Rivadeniera, AF, Jousin, RM, Dellon, AL, Treatment of Peripheral Neuropathy in Leprosy: The case for nerve decompression, Plast Reconstr Surg GO, 2016; 4:e637; doi10.1097/GO0000000000000641.

b) b) Wan, E, Rivadeniera, AF, Serrano, HA, Napit, I, Garbino, JA, Joshua, J, Cardona-Castro, N, Dellon, AL, Theuvenet, W, Protocol for a randomized controlled trial investigating decompression for leprous neuropathy (the DELN protocol), Lepr Rev, 87:553-561, 2016.

Author’s comments: Thanks for your comment. We would like to say that now we understand you and we changed the last paragraph of the DISCUSSION to consider your point. In our conclusion, we considered difficult to include it because we did not consider this surgery procedure in our assistance unfortunately.

Lines 634-42: ….“It is worth emphasizing that those patients who, even after multidrug therapy, still have residual disability detected by the SWM test, may benefit from appropriate peripheral nerve decompression surgery, preventing definitive neural damage from leprosy, as described by Wan et al [34,35].”

Specifically in conclusion, we changed the writing explaining better about the SWM-test as a tactile sensitivity test: 

Line 648 – “The tactile sensation test using Semmes-Weisntein monofilaments proved to be an important instrument…”

Line 673 – “Finally, hand and foot tactile sensation test using Semmes-Weinstein monofilaments proved to be effective…”

Additionally, we received recently (after our submission to the journal) one communication from the Brazilian health Ministry together SORRI, the company that produces the monofilament kits in Brazil about the change of the green monofilament from 0.05g to 0.07g recently adopted the new kit manual. According to the document below signed by Mr. Antony R. J. Nicholl, owner of the company with large expertise in SWM (we have made a free translation for English language), and considering that all our calculations were made considering values greater than 0.05 and as there is no value between 0.05 and 0.07, it has not really changed in our results and analysis, so we decided to do this up to date in our manuscript changing from 0.5 g-force to 0.07 g-force.

In the manuscript we changed it in lines: 123, 136, 151, 159, 162, 167, 172, 180, 200 (Table 1), 255, 271, 290 and 495.

Thank you very much for these so important contributions!

---

## [Decision Letter · Decision Letter 2]

14 Jul 2022

Evaluation of altered patterns of tactile sensation in the diagnosis and monitoring of leprosy using the Semmes-Weinstein monofilaments

PONE-D-21-17910R2

Dear Dr. Frade,

We’re pleased to inform you that your manuscript has been judged scientifically suitable for publication and will be formally accepted for publication once it meets all outstanding technical requirements.

Kind regards,

George Vousden

Staff Editor

PLOS ONE

Additional Editor Comments:

During the revision of your manuscript a reviewer requested that two citations be added to your manuscript, and these have been added as references 34 and 35. Please note that it is not a requirement for publication to cite these references, you may remove them from the version to be typeset if you feel that these works are not relevant to your work.

Reviewers' comments:

Reviewer's Responses to Questions

**Comments to the Author**

1. If the authors have adequately addressed your comments raised in a previous round of review and you feel that this manuscript is now acceptable for publication, you may indicate that here to bypass the “Comments to the Author” section, enter your conflict of interest statement in the “Confidential to Editor” section, and submit your "Accept" recommendation.

Reviewer #2: (No Response)

2. Is the manuscript technically sound, and do the data support the conclusions?

Reviewer #2: (No Response)

3. Has the statistical analysis been performed appropriately and rigorously? 

Reviewer #2: (No Response)

4. Have the authors made all data underlying the findings in their manuscript fully available?

Reviewer #2: (No Response)

5. Is the manuscript presented in an intelligible fashion and written in standard English?

Reviewer #2: (No Response)

6. Review Comments to the Author

Reviewer #2: great job with revisions

all questions answered

I have nothing else to write

I have nothing else to write

7. PLOS authors have the option to publish the peer review history of their article (what does this mean?). If published, this will include your full peer review and any attached files.

Reviewer #2: No

---

## [Editor Report · Acceptance letter]

2 Aug 2022

PONE-D-21-17910R2 

Evaluation of altered patterns of tactile sensation in the diagnosis and monitoring of leprosy using the Semmes-Weinstein monofilaments 

Dear Dr. Frade:

I'm pleased to inform you that your manuscript has been deemed suitable for publication in PLOS ONE. Congratulations! Your manuscript is now with our production department. 

Kind regards, 

on behalf of

Dr. George Vousden 

Staff Editor

PLOS ONE